# Detection and phylogenetic characterization of Jingmen tick virus in *Amblyomma mixtum* ticks from Costa Rica

Tatiana Murillo,[1,2] Luis Enrique Chaves-González,[1,3] Sarah Temmam,[4] Sergio Bermúdez,[5] Eugenia Corrales-Aguilar,[1,2] Victor M. Montenegro,[6] Nolwenn Dheilly,[4] Adriana Troyo[1,2]

**ABSTRACT** Jingmenviruses are a group of segmented flaviviruses detected in arthropods and vertebrates that have attracted growing public health interest due to the recognition of some members as emerging human arboviral pathogens. As part of a study aimed at deciphering the virome of ticks of medical and veterinary importance in Costa Rica, we detected Jingmen tick virus (JMTV) in host-feeding *Amblyomma mixtum* ticks collected from horses. We assembled three complete genome segments and one partial segment from tick pools. Phylogenetic analyses revealed that JMTV from Costa Rica (JMTV Costa Rica) shares a common viral ancestor with JMTV viruses identified in ticks from the Caribbean and Latin America. Two distinct clades of Jingmenviruses were identified in the American continent, suggesting two distinct introductions: one from Europe/Asia and the other from Africa/Asia. Of note, JMTV Costa Rica falls in the same clade as viruses from Europe and Western Asia, including sequences found in humans. Our study constitutes the first detection of JMTV in *Amblyomma mixtum*. This tick species feeds on a wide range of hosts, including wildlife, domestic animals, and frequently parasitizes humans in Central America. Further research involving the detection of active and past infections by JMTV in humans and horses after tick bites is needed to evaluate the risk of spillover in Central America, including Costa Rica.

**IMPORTANCE** Jingmenviruses are flaviviruses detected in arthropods and vertebrates, reported in several countries worldwide. Some members cause disease and infections in humans; therefore, they are considered emergent human arboviruses. In Costa Rica and Central America, there is no information on tick-associated viruses or the role of ticks as putative vectors of viruses. Here, we report the first regional detection of Jingmen tick virus (JMTV) in *Amblyomma mixtum* ticks collected from horses. We assembled three complete and one partial viral segment from tick pools. Phylogenetic analysis revealed that the JMTV detected in Costa Rica is closely related to other detections from Latin America and the Caribbean and is located in the same clade as viruses reported in humans. Additionally, we detected two separate introductions of JMTV to Latin America. To determine whether this JMTV is an emergent arbovirus locally, research on past or active infections in humans is required.

**KEYWORDS** Jingmen tick virus, virome, Ixodida, *Amblyomma*, phylogeny, emergence, arbovirus

Address correspondence to Tatiana Murillo, tatiana.murillocorrales@ucr.ac.cr.

The authors declare no conflict of interest.

See the funding table on p. 11.

Jingmen tick virus (JMTV) is a segmented ssRNA(+) virus, first reported from *Rhipicephalus microplus* ticks in 2014 in Hubei, China (1). Segment 1 encodes for the RNA-dependent RNA polymerase (RdRP), and segment 3 encodes for a non-structural protein. Both segments are genetically similar to the NS5 and NS3 proteins of orthoflaviviruses, allowing the classification of JMTV and JMTV-related viruses in a large group within the *Flaviviridae*, the Jingmenviruses (1–3). Other viruses within this group are

Alongshan virus (ALSV), Mogiana tick virus (MGTV), Rio Preto tick virus (RPTV), Kindia tick virus, Sichuan tick virus, or *Pteropus lylei* Jingmen virus (4–6). Jingmenviruses have been detected in a wide range of hard tick species, in insects such as mosquitoes and fleas, as well as in mammals such as humans, non-human primates, bats, cattle, and rodents (7–9). Jingmenviruses have a worldwide distribution, exemplified by their detection in China, Japan, Russia, Germany, Turkey, Kenya, Cameroon, France, Finland, Italy, Brazil, Lao, Cambodia, French Antilles, Trinidad and Tobago, Colombia, and Mexico (6, 7, 9–16).

In humans, some Jingmenviruses have been associated with tick bites, as exemplified by the concomitant detection of JMTV in patients suffering from Crimean Congo hemorrhagic fever (CCHF) in Kosovo and by the detection of specific antibodies against JMTV in patients with a history of a tick bite in China (17, 18). Of note, JMTV has been detected in tick salivary glands, induces local inflammation at the bite site, and has been detected in both ticks and hosts simultaneously, highlighting its transmission to vertebrate hosts during blood feeding (11, 18). Along with the detection of genomes and antibodies of Jingmenviruses in humans, two of these viruses, JMTV and ALSV, are pathogenic for humans (18, 19). JMTV causes a mild to severe disease characterized by fever, headache, malaise, lymphadenopathy, and the presence of an itchy and painful scar (18). ALSV infection presents with non-specific clinical symptoms, including headache and fever most commonly, as well as fatigue, depression, coma, poor appetite, myalgia, arthralgia, and rash (19). Thus, Jingmenviruses, particularly those phylogenetically related to JMTV and ALSV virus, have been proposed as potential agents of emerging disease in humans and have become the focus of increased research worldwide (20).

In Central America, the presence of several tick-borne bacterial pathogens of humans and domestic animals has been demonstrated (21, 22). For instance, *Rickettsia rickettsii*, the main causative agent of rickettsia spotted fever, and *Rickettsia amblyommatis* have been mainly detected in *Amblyomma mixtum* ticks, which are frequent ectoparasites of humans in this region (23–25). Other tick-borne human pathogens, such as *Rickettsia parkeri*, *Rickettsia africae*, *Anaplasma phagocytophilum*, *Ehrlichia chaffeensis*, *Borrelia burgdorferi* s.l., and relapsing fever *Borrelia* spp., have also been reported in the region in other tick species (21, 22, 26). Since *A. mixtum* has a wide host range, including wildlife, domestic animals, and humans, it is considered a potential vector of pathogens of medical and veterinary importance in Central America, including viruses (24, 25). However, to date, no information is available on tick-associated viruses or the potential risk posed by ticks as vectors of emerging viruses in Central America.

Given the limited information and the potential presence of tick-associated viruses in the region, we investigated the viral communities of ticks from Costa Rica using metagenomic approaches, focusing on human-biting species, including *A. mixtum*. Here, we report the first detection and phylogenetic characterization of a globally emerging virus, Jingmen tick virus, in *A. mixtum* from Central America (JMTV Costa Rica).

## RESULTS

### Identification of JMTV in *A. mixtum* ticks

A total of 97 *A. mixtum* ticks were collected in La Siberia and organized into eight pools (Table 1). Metagenomic sequencing was performed for all pools; however, contigs assigned to JMTV were only detected in pools am04, am05, and am06, corresponding to female (engorged) and male (unengorged and partially engorged) host-feeding adult ticks collected from four horses.

The closest viral genome corresponded to JMTV identified in a pool of *Amblyomma variegatum* and *Rhipicephalus microplus* collected from cattle in the French Antilles (27). The percentage of nucleotide identity and coverage of each segment relative to JMTV French Antilles varies across segments and pools, with the highest coverage and percentage of identity obtained for segments 1 and 2 and the lowest for segments 3 and 4 (Table 2). Nucleotide dissimilarity matrices were calculated to determine differences

**TABLE 1** Summary of tick pool metadata including pool ID, collection date, composition, blood-feeding status, collection method, and JMTV contig count

| Pool ID | Collection date (mo-d-yr) | Composition | Blood-feeding status | Collection method | JMTV contig count |
|---|---|---|---|---|---|
| am_01 | 04-30-2024 | 20 nymphs | Unfed | Flagging | 0 |
| am_02 | 04-30-2024 | 20 nymphs | Unfed | Flagging | 0 |
| am_03 | 04-30-2024 | 20 nymphs | Unfed | Flagging | 0 |
| am_04 | 10-12-2023 | 8 females | Engorged | Direct collection from *Equus caballus* | 268 |
| am_05 | 10-12-2023 | 9 females | Engorged | Direct collection from *Equus caballus* | 128 |
| am_06 | 10-12-2023 | 18 males | Unengorged, partially engorged | Direct collection from *Equus caballus* | 890 |
| am_07 | 04-30-2024 | 1 male | Unfed | Flagging | 0 |
| am_08 | 04-30-2024 | 1 female | Unfed | Flagging | 0 |

between the genome segments obtained from each tick pool. The p-distances obtained were the lowest for segment 1 and the highest for segment 2, in both cases between pools am05 and am06 (Table S1). As p-distances were low between tick pools, indicating low intrapopulation diversity, we combined reads from the three tick pools to obtain a metagenome-assembled genome (MAG) of JMTV Costa Rica. The complete coding sequences of segments 1, 2, and 3 of JMTV Costa Rica were obtained, as well as 60% of segment 4, corresponding to the complete coding sequence of the membrane protein.

## Phylogenetic relationship of JMTV Costa Rica within the diversity of Jingmenviruses

Analyses of the phylogenetic relationships of JMTV Costa Rica within the diversity of Jingmenviruses were conducted for segment 1 (Fig. 1A) and segment 2 (Fig. 1B). Three main clades were observed: clade A consists solely of the JMTV sequence identified in the *Pteropus lylei* bat from Cambodia, and clade B contains the ALSV identified in *Ixodes ricinus* and *I. persulcatus* ticks and in humans, as well as other sequences detected in other *Ixodes* species. JMTV Costa Rica falls in clade C, which includes most of JMTV-related sequences, including MGTV and RPTV from Brazil, Kindia tick virus from Guinea, and Heilongjiang and Sichuan viruses from China.

Clade C is further divided into three subclades, tentatively named CI–CIII. Clade CI comprises only RPTV identified in *Amblyomma sculptum*. Clade CII includes JMTV genomes detected in ticks from Latin America (Costa Rica, Trinidad and Tobago, and

**TABLE 2** Percentage of identity and coverage of consensus sequences obtained for JMTV Costa Rica compared to the reference genome from JMTV French Antilles

| Reference | MN095523 | MN095524 | MN095525 | MN095526 |
|---|---|---|---|---|
| Genome segment | 1 (RdRp) | 2 (glycoprotein) | 3 (NS3-like) | 4 (capsid/membrane) |
| Length (nt) | 3,044 | 2,309 | 2,537 | 2,654 |
| am4 | | | | |
| Consensus length (nt) | 3,042 | 2,279 | 1,938 | 1,869 |
| % identity | 96.3 | 93.1 | 98.1 | 92.6 |
| % coverage | 99.9 | 98.7 | 76.4 | 70.4 |
| am5 | | | | |
| Consensus length (nt) | 2,522 | 2,142 | 1,881 | 433 |
| % identity | 95.8 | 91.2 | 97.6 | 93.5 |
| % coverage | 82.9 | 92.8 | 74.1 | 16.3 |
| am6 | | | | |
| Consensus length (nt) | 2,960 | 2,309 | 2,366 | 1,970 |
| % identity | 96.5 | 93.2 | 97.7 | 91.7 |
| % coverage | 97.2 | 100.0 | 93.3 | 74.2 |

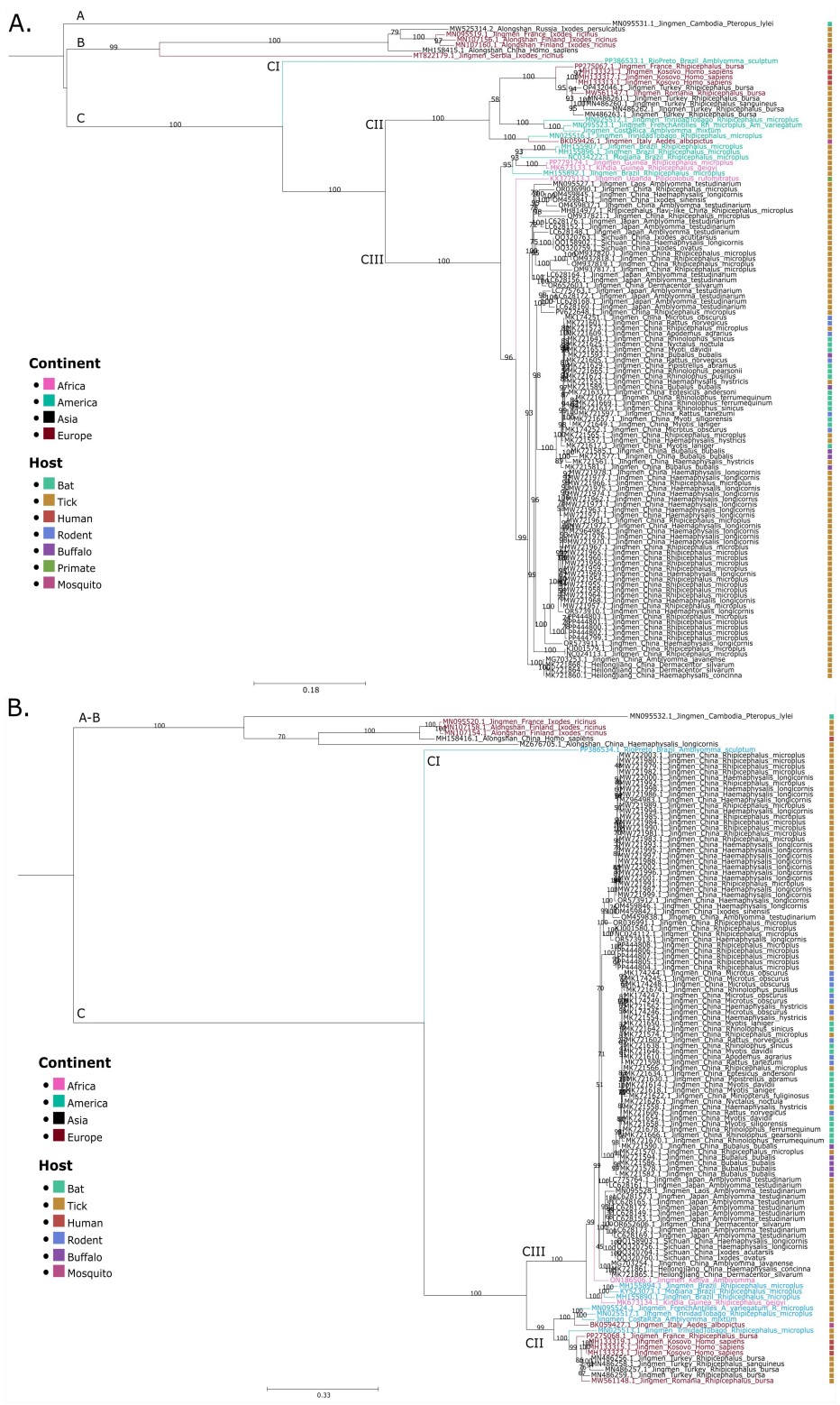

**FIG 1** Phylogenetic characterization of JMTV Costa Rica within the Jingmenviruses. Phylogenetic trees of (A) segment 1, encoding for the RdRp and (B) segment 2, encoding for the glycoprotein. MAGs from JMTV Costa Rica were assembled in CLC Genomics from three tick pools collected from the same collection site. A total of 123 complete sequences for each segment were downloaded from the GenBank database. Alignments were performed with only the open reading frame in MAFFT and trimmed in MEGA 12. Trees were calculated with IQ-TREE and ModelFinder (best fit model for each segment: [i] GTR + F + (Continued on next page)

Fig 1 (Continued)

R3 and [ii] TIM2 + F + G4). Bootstraps were estimated with 1,000 replicates and approximate Bayes test and are shown in each branch. Midpoint rooted trees were visualized in TreeViewer and Inkscape. Sequences are color-coded by country of origin and host, as indicated in the legend. Clades are marked with letters A–C. Subclades are indicated from CI to CIII.

French Antilles) and Europe/Western Asia (France, Turkey, and Romania), an *Aedes albopictus* from Italy, and JMTV genomes detected in patients suffering from CCHF in Kosovo (17, 28). Finally, most of the JMTV sequences belong to clade CIII, encompassing genomes mainly identified in ticks but also in primates, rodents, buffaloes, and bats from Asia, Africa, and Brazil.

Interestingly, both the phylogenetic and pairwise distance analyses revealed that the closest genome of JMTV Costa Rica varies across the segment considered (Fig. 2; Tables S2 to S5). For segment 1 and segment 3, JMTV Costa Rica is more closely related to JMTV French Antilles detected in a pool of *Rhipicephalus microplus* and *Amblyomma variegatum*, followed by JMTV Trinidad and Tobago detected in *R. microplus*. Conversely, for segments 2 and 4, JMTV Costa Rica is more closely related to JMTV Trinidad and Tobago obtained from *R. microplus*, followed by JMTV French Antilles, detected in a pool of *R. microplus* and *A. variegatum* (7, 29). Thus, it can be hypothesized that JMTV Costa Rica may have originated from a combination of segments from JMTV French Antilles and JMTV Trinidad and Tobago.

## Phylogenetic relationship of JMTV Costa Rica to other detections from Latin America

JMTV has been detected in other Latin American countries, such as in ticks from Mexico and Colombia, but only partial genome segments are available (15, 16). To analyze the phylogenetic relationship of JMTV Costa Rica to these JMTVs from Latin America, we conducted two additional analyses based on partial segment 1 (including JMTV Mexico, Fig. S1A), and partial segment 2 (containing JMTV Colombia, Fig. S1B). Results showed that JMTV Mexico belongs to subclade CII, like JMTV Costa Rica, and is closely related to strains from the French Antilles and Trinidad and Tobago. JMTV Colombia also belongs to subclade CII but is more closely related to a virus detected in *Aedes albopictus* from Italy. Of note, Brazilian Jingmenviruses (JMTV and MGTV) fall into a distinct subclade CIII (Fig. 1; Fig. S1) compared to other Latin American genomes (Costa Rica, French Antilles, Mexico, Colombia, and Trinidad and Tobago), which are in subclade CII, highlighting two different ancestors and possibly two different introductions of JMTV in the continent.

## DISCUSSION

As a result of an ongoing investigation of the virome of *A. mixtum* ticks, we report the first detection of JMTV in Central America, a virus considered a global emerging pathogen since it has been reported in multiple countries, detected in various hosts, including arthropods and animals, and notably associated with human diseases (7, 18, 20). We obtained the complete coding sequences of viral segments 1–3 (coding for the RdRP, glycoprotein, and NS3, respectively) and the complete coding sequence of the membrane protein from segment 4. We demonstrated that the closest phylogenetic relatives of JMTV Costa Rica are JMTV genomes from the Caribbean, Latin America, Europe, and Western Asia.

JMTV has been detected in other *Amblyomma* species, such as *Amblyomma javanense* (China), *Amblyomma testudinarium* (Japan, Laos, and China), *Amblyomma dissimile* (Colombia), *Amblyomma variegatum* (French Antilles, although in this case the origin might be from *R. microplus* as the authors could not establish it), and an unidentified *Amblyomma* species (Kenya) (4, 7, 13, 18, 30). Here, we report the first detection of JMTV in the *Amblyomma cajennense* species complex, specifically in *A. mixtum*. This species ranges from the southern United States to Ecuador and Colombia, including some Caribbean islands like Trinidad and Tobago and Jamaica (31). Like other species

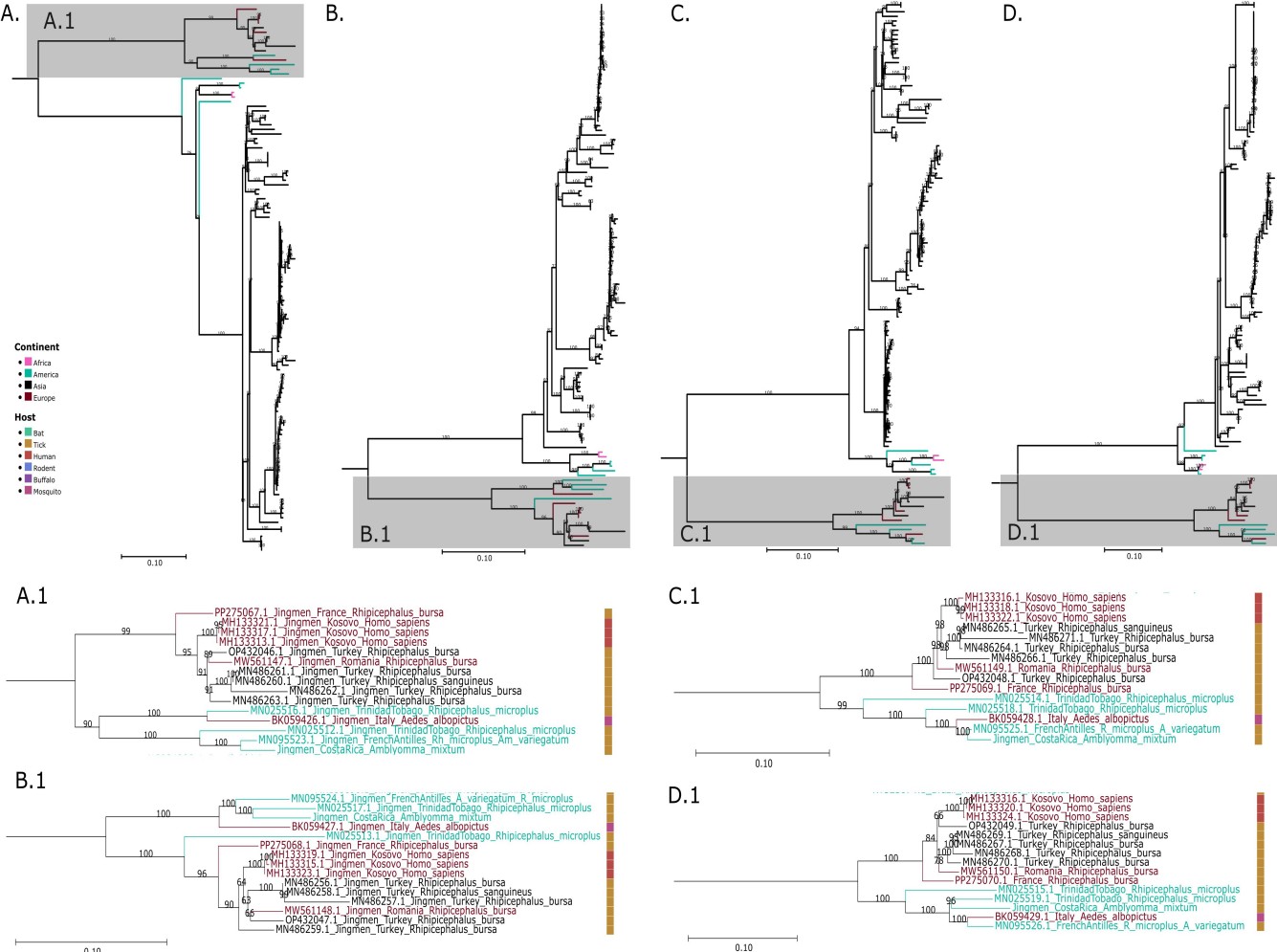

**FIG 2** Phylogenetic characterization of JMTV Costa Rica compared with Jingmenviruses belonging to the same clade. Phylogenetic trees of (A) segment 1 (RdRp), (B) segment 2 (glycoprotein), (C) segment 3 (NS-3 like protein), and (D) segment 4 (membrane protein). Cropped regions from the subclade CII are shown in A.1. (segment 1), B.1. (segment 2), C.1. (segment 3), and D.1. (segment 4). MAGs from JMTV Costa Rica were assembled in CLC Genomics from three tick pools collected from the same collection site. A total of 117 complete sequences for each segment were downloaded from the GenBank database. Alignments were performed with only the open reading frame in MAFFT and trimmed in MEGA 12. Trees were calculated with IQTree and ModelFinder (best fit model for each segment: [1] GTR + F + R3, [2] TIM3 + F + R3, [3] GTR + F + I + G4, and [4] TIM2 + F + I + G4). Bootstraps were estimated with 1,000 replicates and approximate Bayes test and are shown in each branch. Midpoint rooted trees were visualized in TreeViewer and Inkscape. Sequences are color-coded by country of origin and host, as indicated in the legend.

in the complex, *A. mixtum* often bites humans and is the most common tick parasitizing humans in Central America (25). Because it has an extensive host range, including wild and domestic mammals, *A. mixtum* ticks may play a key role in pathogen transmission across vertebrates, including *R. rickettsii* in Central America (24, 25). Its role in the transmission of arboviruses is currently unknown, but its feeding habits and high abundance compared to other *Amblyomma* species in human-modified environments highlight its potential as a vector for emerging viral zoonotic pathogens (22, 32). Therefore, detecting JMTV in *A. mixtum* could indicate a risk of human exposure to this virus and warrants further investigation.

In this study, JMTV was only detected in pools of *A. mixtum* ticks collected from horses and not from questing ticks of the same species. The area where questing and host-feeding ticks were collected are pastures for cattle grazing with horses. These areas are connected to crop plantations and highly diverse forests, enabling the movement of wildlife species within the region. Consequently, there are many potential hosts for

*A. mixtum* and other tick species, including cattle, dogs, non-human primates, rodents, small mammals, birds, and bats, which could serve as JMTV reservoirs. Among those previously mentioned, JMTV has been detected in cows, non-human primates, rodents, and bats (7, 8, 10, 11, 33). Because all the JMTV-positive pools included ticks with equid blood at the time of collection, JMTV detection in the pool could be attributed to tick infection, tick cofeeding, horse infection, horse viremia, or a combination of these; however, to our knowledge, JMTV has not been detected in horses (6, 8). It is now necessary to test for active JMTV infection in horse samples using molecular methods or virus isolation and to use serological analyses to assess past JMTV infections and eventually determine if JMTV Costa Rica is transmitted to horses. However, we should mention that infectivity and replication of JMTV in the ticks were not assessed in this study to be able to determine if these ticks could have transmitted JMTV to the horses. Further studies should also consider the role of other arthropod vectors in the circulation of JMTV in Costa Rica. In addition to screening other tick species for JMTV, mosquitoes may also be considered, given that the closely related JMTV from Italy was detected in *Ae. albopictus* mosquitoes (28).

Phylogenetic analyses showed that JMTV Costa Rica clusters in clade C comprising other genomes detected in Latin America and the Caribbean (French Antilles, Trinidad and Tobago, Colombia, and Mexico) and from Western Europe to Southwestern Asia (France, Kosovo, Romania, Italy, and Turkey) (7, 15–17, 28, 29, 33, 34). This clade has been called the mid-latitude lineage because a correlation between the latitudinal distribution of these viruses and their genetic diversity has been observed (6). The detection of JMTV Costa Rica in La Siberia is consistent with the mid-latitude detection of JMTV but not with its detection in *Amblyomma*, as the authors propose that mid-low latitude JMTV is predominantly detected in *Amblyomma* (6). This is also the case for the reports from Colombia (*A. dissimile*) and possibly from the French Antilles (*A. variegatum*) (15, 27). Further virome studies of autochthonous tick species from Latin America are required to determine this proposed latitudinal distribution pattern of tick-associated JMTV.

Within clade C, two distinct subclades were detected in the American continent. JMTV viruses detected in Costa Rica, Mexico, Colombia, and the Caribbean (French Antilles and Trinidad and Tobago) belong to subclade CII and are phylogenetically related to viruses from Western Europe to Southwestern Asia. In contrast, viruses found in Brazil belong to subclade CIII, which is more closely related to viruses from Africa (Guinea) and Southeast Asia. So far, only the migration of the virus from Trinidad and Tobago to the French Antilles and then to Italy has been reported (6). Our results suggest that JMTV in America has two distinct origins, suggesting two independent introductions in the continent: one from Europe/Southwestern Asia and one from Africa/Southeast Asia. The origin of these introductions (movement of wild or domestic animals, arthropods, and/or humans) warrants further investigation.

Comparisons of the genetic similarities between JMTV Costa Rica and its closest viral relatives showed that the most similar genomes slightly varied, depending on the segment considered, but remained in the same subclade: for segments 1 and 3, JMTV Costa Rica was closer to JMTV French Antilles, while for segments 2 and 4, JMTV Costa Rica was closer to JMTV Trinidad and Tobago rather than JMTV French Antilles (7, 29). These results suggest that reassortment events may have occurred between JMTV French Antilles and JMTV Trinidad and Tobago, leading to the emergence of JMTV Costa Rica. However, additional samplings and JMTV detections in Central and Latin America are needed to confirm this hypothesis that is based on segment-specific phylogenetic incongruence. For a reassortment to occur, a host (arthropod or vertebrate) must be simultaneously infected by two viral strains, and segment exchange may occur in cells coinfected with the two strains. Reassortment may enhance viral fitness by providing advantages for adaptation to new hosts and immune escape (35). These events have been previously proposed for segments 2 and 4 in other Jingmenviruses (4). The putative reassortment origin of JMTV Costa Rica needs further investigation.

This is the first detection of JMTV in Central America, which complements previous findings from the Caribbean, Mexico, and South America (4, 7, 11, 15, 29). We nearly obtained the first complete genome; only the capsid gene remains incomplete. However, this was only possible by combining contigs from three different tick pools, and we recognize that this approach might obscure differences in nucleotide sequences and intrapopulation diversity. Nonetheless, such issues can also occur when working with tick pools, as it is impossible to determine the exact viral sequence coming from each tick. We can only be certain about the viral genome from viral isolates; therefore, future efforts should also attempt virus isolation from ticks.

In dengue-endemic countries such as Costa Rica, during outbreaks, clinical diagnosis of dengue is performed without laboratory confirmation (36). This can lead to other diseases being misdiagnosed as dengue, as the clinical presentation is difficult to distinguish from other fever-producing illnesses (37, 38). Most dengue-endemic countries are located in hotspots of pathogen emergence, and such clinical diagnosis could prevent the detection of other emerging infections in humans (39, 40). Studies like ours will aid in detecting emerging viruses in these regions. Indeed, although JMTV Costa Rica was not phylogenetically related to the recognized human pathogen ALSV circulating in China, it was phylogenetically related to JMTV detected in patients with CCHF in Kosovo (17–19). However, to establish whether JMTV Costa Rica is able to infect vertebrates and if it could be a human pathogen, future steps should test humans with a tick bite for active infections or conduct serological tests on people in the region where JMTV was detected.

The detection of JMTV in Costa Rica highlights the need for further research to determine whether the virus is infecting humans and animals and potentially causing disease. This requires identifying active infections and/or conducting serological testing to assess prior exposure. Additionally, it is crucial to investigate whether *A. mixtum* is infected and capable of transmitting JMTV to its vertebrate hosts, given its high potential as a vector of human diseases. Our study represents a pioneering effort in the investigation of tick-associated viruses in a region recognized as a potential hotspot for pathogen emergence, employing cutting-edge technology to advance this critical area of research.

## MATERIALS AND METHODS

### Specimen collection

Ticks were collected in La Siberia, located at approximately 18 m.a.s.l. in the Valle de La Estrella district, Limón Province (9°45′38.4″ N, 82°55′33.8″ W), in the Atlantic coast of Costa Rica, between October 2023 and May 2024. This site was selected based on preliminary sampling that confirmed the presence of *A. mixtum* in the locality. The biotope at the sampling site was a grassland parcel of broken terrain for grazing, subject to trampling by productive species, and surrounded by an intervened forest dominated by mature trees with sparse canopies and wide spacing between trees. It is inhabited by humans, cattle, horses, and other domestic and companion animals, all of which are suitable hosts for *A. mixtum* ticks. Questing ticks were collected by flagging the pasture where horses were foraging, while unengorged, partially engorged, and fully engorged host-feeding ticks were directly collected from horses (*Equus caballus*) with forceps. Upon collection, specimens were kept alive and transported in vials containing cotton moistened with distilled water.

### Specimen identification

Live ticks were transported to the Medical Entomology Laboratory at the University of Costa Rica for identification. Morphological identification was carried out, employing available dichotomous keys for regional species (41–43). Once identified, specimens were frozen at −80°C for further processing. Molecular species confirmation was performed by amplification of the cytochrome *c* oxidase subunit I (*cox-1*) gene,

as previously described (44). Amplicons were purified with ExoSAP-IT (Thermo Fisher Scientific, USA) according to the manufacturer's instructions and sequenced at Macrogen Inc. (South Korea). Sanger sequences were trimmed in BioEdit (45) and compared to the database of the National Center for Biotechnology Information (NCBI) with BLASTn (46).

## RNA extraction

Ticks were washed twice in 70% ethanol and once in RNase-free water, then adult ticks were cut lengthwise with a sterile scalpel. Individual halves were stored at −80°C. The other halves were pooled before RNA extraction. Pools were constituted as follows: adult ticks were pooled by species, collection method (flagging or direct collection from horses), stage, and sex, rendering five pools of 1–18 ticks each. Nymphs were pooled into groups of 20 individuals to create three additional pools (Table S1). Tick pools were homogenized with a sterile pestle in 200 µL RNAlater (Invitrogen, USA) in two 30-s intervals with 1 min resting on ice between intervals. RNA extraction was performed using the RNeasy Mini Kit (Qiagen, Germany) according to the manufacturer's instructions. After extraction, 1 µL of Ribolock RNase Inhibitor 40 U/µL was added to extracted RNA, followed by a DNAse treatment using the DNA-free Kit (Invitrogen) to reduce remnant genomic DNA. The extracted RNA and the remnant genomic DNA were quantified using the QuantiFluor RNA System and QuantiFluor dsDNA System kits (Promega, USA). To further confirm the absence of amplifiable genomic DNA after DNase treatment, a *cox-1* PCR was performed as previously described, using the extracted RNA as a template (44).

## Reverse transcription

RNA reverse transcription was carried out using the Maxima H Minus First Strand cDNA Synthesis Kit (Thermo Fisher Scientific). To ensure the presence of amplifiable cDNA, a PCR for *cox-1* was performed as previously described, using the reverse transcription product as a template (44). cDNA was quantified as described above and stored at −20°C until further processing.

## Next generation sequencing

Sequenase v2.0 DNA Polymerase kit (Thermo Fisher Scientific) was used to synthesize the cDNA complementary strand. Then, a random amplification was performed using the MALBAC Single Cell Whole Genome Amplification kit (Proteigene, China), as previously described (47). Libraries were prepared starting from the double-stranded cDNA using the Illumina DNA Prep Kit (Illumina, USA) according to the manufacturer's instructions. Sequencing was performed on the Illumina NextSeq 2000 platform, set to 2 × 100 bp read length with paired-end indexes on a P2 flowcell (Illumina), yielding approximately 50 Mi paired reads per library.

## Virus taxonomic assignation

Raw reads were processed using Microseek (48). This analysis pipeline includes read quality checks, trimming, and normalization, followed by *de novo* assembly, open reading frame prediction, and contig and singleton taxonomic assignment using specialized (RBDV-prot) and generalist (NCBI/nr and NCBI/nt) databases (49).

## JMTV Costa Rica metagenome assembly and completion

Contigs and reads classified as JMTV were mapped against the four JMTV segments from French Antilles (MN095523.1, MN095524.1, MN095525.1, and MN095526.1) using default parameters in the CLC Genomics Workbench v23.0.5 (Qiagen). For MAG assembly, p-distances were calculated in MEGA 12 (50) to determine nucleotide differences between segments obtained from each pool (am04, am05, and am06). Complete consensus sequences were obtained for segments 1 and 2 and partial sequences for

segments 3 and 4. Only the membrane gene sequence was completed for segment 4. Segment 3 was completed using PCR amplification and Sanger sequencing. PCR was performed in a total volume of 25 µL using primers JMTV-CR-S3-For (5′-GCGTCAGACTC ACCAAACAG-3′) and JMTV-CR-S3-Rev (5′-CTCGGTATATCCCCTCTGCT-3′) and the Phusion Taq polymerase (New England Biolabs, USA) under the following conditions: 5 µL of 5× HF buffer, 1 µL of each 10 µM primer, 0.5 µL of Phusion, and 5 µL of mixed cDNA (combination of cDNA from am04, am05, and am06 samples in equal volume). Cycling conditions were denaturation at 98°C for 30 s, followed by 45 cycles of amplification (98°C for 10 s, 55°C for 30 s, and 72°C for 1 min), and a final extension at 72°C for 7 min. TAE 2% agarose gel electrophoresis was performed to verify the amplification. An amplicon of the correct size (598 bp) was extracted from the gel and purified using the NucleoSpin Gel and PCR Clean-Up Kit (Macherey-Nagel, USA) according to the manufacturer's recommendations. Sanger sequencing was outsourced to Eurofins Genomics. Sanger sequences were aligned with the partial consensus sequence of segment 3 to obtain the full coding sequence of the segment.

## Phylogenetic analysis and nucleotide distance estimations

A total of 123 complete genomes of Jingmenviruses were obtained from the GenBank NCBI database (Table S1) (46). Partial sequences from JMTV detected in Mexico and Colombia were included in the analysis to enrich the data set of Latin American sequences. Phylogenetic reconstructions were conducted as follows: (i) based on the complete genome segments 1 and 2 of all Jingmenviruses, (ii) based on the complete genome of JMTV virus species, and (iii) based on the partial segments 1 and 2 of JMTV detected in Latin America. All alignments were performed using MAAFT v7 (FFT-NS-1 parameter) and manually checked and trimmed in MEGA12 (50, 51). Phylogenetic trees were calculated in IQTree using ModelFinder, and branch support was calculated with ultrafast bootstrap (1,000 replicates) and the single-branch test with the Approximate Bayes test (52–54). The best-fit model, which varies between phylogenetic trees, is provided in the figure legends. Sequences from the closest phylogenetic relatives of the four JMTV Costa Rica segments were used to calculate nucleotide pairwise distance matrices using the Maximum Composite Likelihood method in MEGA12 (50).

## ACKNOWLEDGMENTS

Sampling and sample processing were funded by grant 803-C3-463 from the Vice Directory of Research and the Office of International Affairs and External Cooperation, both from the University of Costa Rica. Sequencing and data analyses were funded by Institut Pasteur and by the National Institute of Allergy and Infectious Diseases of the National Institutes of Health under Award Number U01AI151758 to N.D. The content is solely the responsibility of the authors and does not necessarily represent the official views of the National Institutes of Health. This manuscript was supported by PUBLICARE-UCR and Open Access funding was enabled and organized by CIHRED-UCR.

We are grateful to the people of La Siberia in Valle de la Estrella for allowing us to collect ticks from their horses and paddocks and to Erick López Sánchez for his technical assistance during fieldwork.

T.M., A.T., and L.E.C.-G. designed the study. T.M., A.T., and N.D. secured funding. L.E.C.-G., V.M.M., A.T., S.B., and T.M. conducted the sample collection, field data gathering, and tick identification. L.E.C.-G., T.M., N.D., S.T., and E.C.-A. handled sample processing, methodology, sequencing, and data analyses. T.M. carried out the final data analysis and visualization and drafted the initial version of the manuscript. L.E.C.-G., S.T., and A.T. contributed to writing the initial draft. All authors interpreted the results and revised the manuscript.

## AUTHOR AFFILIATIONS

[1]Centro de Investigación en Enfermedades Tropicales, Universidad de Costa Rica, San José, Costa Rica

[2]Sección de Virología, Facultad de Microbiología, Universidad de Costa Rica, San José, Costa Rica

[3]Sección de Entomología Médica, Facultad de Microbiología, Universidad de Costa Rica, San José, Costa Rica

[4]Pathogen Discovery Laboratory, Institut Pasteur, Paris, France

[5]Departamento de Investigación en Entomología Médica, Instituto Conmemorativo Gorgas de Estudios de la Salud, Ciudad de Panamá, Panamá

[6]Laboratorio de Parasitología, Escuela de Medicina Veterinaria, Universidad Nacional, Heredia, Costa Rica

## AUTHOR ORCIDs

Tatiana Murillo  http://orcid.org/0000-0001-6346-9509
Luis Enrique Chaves-González  http://orcid.org/0000-0003-1201-3979
Sarah Temmam  http://orcid.org/0000-0003-3655-9220
Sergio Bermúdez  http://orcid.org/0000-0003-1830-3133
Eugenia Corrales-Aguilar  http://orcid.org/0000-0002-0845-8329
Victor M. Montenegro  https://orcid.org/0000-0002-0226-5155
Nolwenn Dheilly  http://orcid.org/0000-0002-3675-5013
Adriana Troyo  http://orcid.org/0000-0001-9513-9969

## FUNDING

| Funder | Grant(s) | Author(s) |
|---|---|---|
| National Institutes of Health | U01AI151758 | Nolwenn Dheilly |
| Universidad de Costa Rica | 803-C3-463 | Adriana Troyo |

## AUTHOR CONTRIBUTIONS

Tatiana Murillo, Conceptualization, Data curation, Formal analysis, Funding acquisition, Investigation, Methodology, Resources, Software, Supervision, Validation, Visualization, Writing – original draft, Writing – review and editing | Luis Enrique Chaves-González, Conceptualization, Data curation, Formal analysis, Funding acquisition, Investigation, Methodology, Validation, Writing – original draft, Writing – review and editing | Sarah Temmam, Conceptualization, Data curation, Formal analysis, Investigation, Methodology, Project administration, Resources, Software, Supervision, Validation, Visualization, Writing – original draft, Writing – review and editing | Sergio Bermúdez, Conceptualization, Investigation, Methodology, Resources, Supervision, Writing – review and editing | Eugenia Corrales-Aguilar, Conceptualization, Investigation, Methodology, Validation, Writing – review and editing | Victor M. Montenegro, Conceptualization, Investigation, Methodology, Resources, Supervision, Writing – review and editing | Nolwenn Dheilly, Data curation, Formal analysis, Funding acquisition, Investigation, Methodology, Project administration, Resources, Software, Supervision, Validation, Writing – review and editing | Adriana Troyo, Conceptualization, Data curation, Formal analysis, Funding acquisition, Investigation, Methodology, Project administration, Resources, Supervision, Writing – original draft, Writing – review and editing

## DATA AVAILABILITY

Complete and partial genome segments were deposited in GenBank under accession numbers PX635876 to PX635879.

## ETHICS APPROVAL

This project was approved by the Institutional Biodiversity Committee (CBio) from the University of Costa Rica in CBio-82-2022, resolution 375.

## ADDITIONAL FILES

The following material is available online.

### Supplemental Material

**Supplemental material (Spectrum04078-25-s0001.pdf).** Fig. S1; Tables S1 to S5.

### Open Peer Review

**PEER REVIEW HISTORY (review-history.pdf).** An accounting of the reviewer comments and feedback.

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
