## [Reviewer comments · Microbiology Spectrum]

Microbiology Spectrum

Detection and phylogenetic characterization of Jingmen tick virus in *Amblyomma mixtum* ticks from Costa Rica

Tatiana Murillo, Luis Chaves-González, Sarah Temmam, Sergio Bermúdez, Eugenia Corrales-Aguilar, Victor Montenegro, Nolwenn Dheilly, and Adriana Troyo

Corresponding Author(s): Tatiana Murillo, Universidad de Costa Rica

Review Timeline:

Submission Date:	December 16, 2025
Editorial Decision:	February 4, 2026
Revision Received:	March 9, 2026
Accepted:	March 29, 2026

Editor: Day-Yu Chao

Reviewer(s): The reviewers have opted to remain anonymous.

Transaction Report:

DOI: <https://doi.org/10.1128/spectrum.04078-25>

Re: Spectrum04078-25 (Detection and phylogenetic characterization of Jingmen tick virus in *Amblyomma mixtum* ticks from Costa Rica)

Dear Ms. Tatiana Murillo:

Thank you for the privilege of reviewing your work. Below you will find my comments, instructions from the Spectrum editorial office, and the reviewer comments.

Revision Guidelines

Sincerely,
Day-Yu Chao
Editor
Microbiology Spectrum

Reviewer #1 (Comments for the Author):

In this manuscript, Murillo et al. presents the first detection of Jingmen tick virus (JMTV) in Central America in the ticks collected from horses in Costa Rica. The authors combines metagenomic sequencing, genome assembly, and phylogenetic analyses to situate the Costa Rican strain within the global diversity of Jingmenviruses.

The strengths of the paper includes the first reporting of JMTV in Central America and the first association with *Amblyomma mixtum*, an ecologically and epidemiologically important human-biting tick species in the region, hence the public health relevance is justified. The methods utilise robust sequencing and phylogenetic analysis. The work describes reassortment on the basis of segment-specific phylogenetic incongruence supported by pair-wise distance matrices. Proposed hypothesis of multiple introductions of JMTV into the Americas is plausible and aligns with recent literature.

The work is timely, geographically novel, and appears relevant to the recently growing literature on emerging tick-borne viruses with zoonotic potential. The study is well executed, the analyses appear technically sound, and is clearly written and presented. With a few clarifications and some cautious related to framing of certain conclusions, this work would make a good contribution to the field. Here are some concerns

Major concerns

1. The interpretation of reassortment is based on segment specific phylogenetic incongruence and hence is more suggestive than definitive. Hence, the claims should be rephrased to outline this limitation and explicitly state that it is a hypothesis.
2. JMTV was detected only in engorged ticks collected from horses. While the authors discuss multiple possibilities (tick infection, horse viremia, co-feeding), the Results and Discussion occasionally blur the distinction between vector infection and bloodmeal detection. More explicitly distinguish between detection of viral RNA and evidence of active tick infection. Consider adding a short clarifying statement in the Results section emphasizing that infectivity and replication were not assessed.
3. As authors combined 3 pools to generate a single MAG, this strategy may include inadvertently intra-population diversity. Authors should consider commenting on whether any segment-specific SNPs or minor variants were observed between pools, and how much the segments were identical to each other.
4. The authors suggestion of zoonotic risk based purely of phylogenetic relatedness is a bit of an overstatement in light of lack of any demonstrated pathogenicity. This could be mentioned cautiously

Minor concerns

1. Please used consistently "Jingmenvirus group" or "Jingmenviruses" throughout the text.
2. Phylogenetic trees are generally clear, but some labels in supplementary figures are small and difficult to read. I had hard time reading clearly these minutest of words.
3. Discussion is slightly long and appears sometimes as repetition for public health implications which could be sharpened for clarity.

Reviewer #2 (Public repository details (Required)):

The manuscript should include a data availability section to describe sequence accession numbers, etc.

Reviewer #2 (Comments for the Author):

Here Murillo et al describe the first detection of Jingmen tick virus in *Amblyomma mixtum* collected from horses in Costa Rica. Virus was detected using pathogen agnostic sequencing approaches and phylogenetics was performed with available Jingmenviruses from NCBI Genbank. Overall, this is a relatively straightforward study that includes useful information, however. I believe there are a few scientific issues and several things that need to be addressed to facilitate understanding by the reader. Specific comments follow.

Line 25: Jingmenvirus should be Jingmenviruses.

Line 106-107 and table 1: more details should be included on tick collections. For example, were all adult females collected off of horses engorged? Or were flat ticks collected as well.

Line 107: what platform was used for metagenomic sequencing?

Line 107-109: what controls were included for metagenomic sequencing? And how did you rule out potential cross-contamination between samples and sequencing libraries?

Line 108: what criteria were used to filter reads?

Line 145-147: Did you perform some kind of recombination test on your sequences, preferably multiple different methods?

Line 166: "infecting" is doing some heavy lifting here. To my knowledge, evidence of "infection" with Jingmenviruses is scant however, there have been frequent "detections" of Jingmenviruses in multiple different species, mostly using molecular methods.

Response to reviewers

1. Reviewer #1:

Comment: *In this manuscript, Murillo et al. presents the first detection of Jingmen tick virus (JMTV) in Central America in the ticks collected from horses in Costa Rica. The authors combines metagenomic sequencing, genome assembly, and phylogenetic analyses to situate the Costa Rican strain within the global diversity of Jingmenviruses.*

*The strengths of the paper includes the first reporting of JMTV in Central America and the first association with *Amblyomma mixtum*, an ecologically and epidemiologically important human-biting tick species in the region, hence the public health relevance is justified. The methods utilise robust sequencing and phylogenetic analysis. The work describes reassortment on the basis of segment-specific phylogenetic incongruence supported by pair-wise distance matrices. Proposed hypothesis of multiple introductions of JMTV into the Americas is plausible and aligns with recent literature.*

The work is timely, geographically novel, and appears relevant to the recently growing literature on emerging tick-borne viruses with zoonotic potential. The study is well executed, the analyses appear technically sound, and is clearly written and presented. With a few clarifications and some cautious related to framing of certain conclusions, this work would make a good contribution to the field. Here are some concerns

Response from authors: We thank the reviewer for their initial comment, which confirms the relevance of this report.

Major concerns:

1. *The interpretation of reassortment is based on segment specific phylogenetic incongruence and hence is more suggestive than definitive. Hence, the claims should be rephrased to outline this limitation and explicitly state that it is a hypothesis.*

Response from authors: Claims about reassortment were rephrased across the manuscript, and when mentioned, it was stated as only a hypothesis.

2. *JMTV was detected only in engorged ticks collected from horses. While the authors discuss multiple possibilities (tick infection, horse viremia, co-feeding), the Results and Discussion occasionally blur the distinction between vector infection and bloodmeal detection. More explicitly distinguish between*

detection of viral RNA and evidence of active tick infection. Consider adding a short clarifying statement in the Results section emphasizing that infectivity and replication were not assessed.

Response from authors: The results and discussion sections were reviewed and edited to remove ambiguous terms related to virus detection vs. infection/replication (e.g., “associated with” and similar terms). In addition, in the discussion section (lines 219-221), we acknowledged that the infectivity and replication of JMTV in ticks were not investigated in this study.

3. *As authors combined 3 pools to generate a single MAG, this strategy may include inadvertently intrapopulation diversity. Authors should consider commenting on whether any segment-specific SNPs or minor variants were observed between pools, and how much the segments were identical to each other.*

Response from authors: We acknowledge that generating a single MAG may include intrapopulation diversity inadvertently. Therefore, before generating each MAG, we mapped each pool independently onto a JMTV reference genome and aligned the 3 consensus sequences to produce a p-distance matrix, which we used to estimate the number of nucleotide differences per site. The p-distances varied between 0.0008 and 0.003 (table 1), thus showing that the number of nucleotide differences between consensus sequences were few in number and therefore they could be combined to produce the final MAG.

Table 1. Nucleotide pairwise distance matrices comparing the consensus sequences obtained for each genome segment in each tick pool.

		am04	am05	am06
Segment 1 Consensus length 2472 nt	am04			
	am05	0,000815		
	am06	0,000809	0,000407	
Segment 2 Consensus length 2025 nt	am04			
	am05	0,002		
	am06	0,001	0,003	
Segment 3 Consensus length 1881 nt	am04			
	am05	0,001		
	am06	0,000	0,001	
Segment 4 Consensus length 426 nt	am04			
	am05	0,002		
	am06	0,000	0,002	

To address the reviewer’s comments, the results were included in a new Supplementary Table 2, and the

text was further extended in the results section through lines 118-123. Additionally, the possibility of inadvertently intra-population diversity was mentioned in the discussion in line 272. Detailed methods for the calculations were addressed in lines 365-367 of the materials and methods section.

4. *The authors suggestion of zoonotic risk based purely of phylogenetic relatedness is a bit of an overstatement in light of lack of any demonstrated pathogenicity. This could be mentioned cautiously.*

Response from authors: The phrase "...which further indicates a risk for human transmission" was eliminated from that sentence and the following sentence was edited to indicate that further studies are needed to evaluate the risk for human infection. In addition, these sentences were transferred to the last part of the discussion, for clarity and to eliminate repetition.

Minor concerns:

1. *Please use consistently "Jingmenvirus group" or "Jingmenviruses" throughout the text.*

Response from authors: The terms were changed in the text and used consistently as suggested.

2. *Phylogenetic trees are generally clear, but some labels in supplementary figures are small and difficult to read. I had hard time reading clearly these minutest of words.*

Response from authors: The figure was improved to increase readability.

3. *Discussion is slightly long and appears sometimes as repetition for public health implications which could be sharpened for clarity.*

Response from authors: The discussion was shortened and repetitions removed to make the public health implications clearer.

2. Reviewer #2:

Comment: Here Murillo et al describe the first detection of Jingmen tick virus in Ambloymma mixtum collected from horses in Costa Rica. Virus was detected using pathogen agnostic sequencing approaches and phylogenetics was performed with available Jingmenviruses from NCBI Genbank. Overall, this is a relatively straightforward study that includes useful information, however. I believe there are a few scientific issues and several things that need to be addressed to facilitate understanding by the reader. Specific comments follow.

Response from authors: We thank the reviewer for their initial comment, which confirms the usefulness of the information reported.

Specific comments:

1. *The manuscript should include a data availability section to describe sequence accession numbers, etc.*

Response from authors: A data availability section was added to the Materials and Methods section (lines 403-405), including sequence deposition and accession numbers, which were previously provided in the section "JMTV Costa Rica metagenome assembly and completion" (lines 385-386).

2. *Line 25: Jingmenvirus should be Jingmenviruses.*

Response from authors: The term was changed in the text and used consistently.

3. *Line 106-107 and table 1: more details should be included on tick collections. For example, were all adult females collected off of horses engorged? Or were flat ticks collected as well?*

Response from authors: We included a column in Table 1 to indicate the blood-feeding status of ticks in each pool. More detail as to engorged ticks, and the results section in line 111 now states: "...contigs assigned to JMTV were only detected in pools am04, am05, and am06, corresponding to female (engorged) and male (unengorged and partially engorged) host-feeding adult ticks...".

4. *Line 107: what platform was used for metagenomic sequencing?.*

Response from authors: Metagenomic sequencing was performed on the Illumina NextSeq 2000 platform as stated in lines 353-355 from the Materials and Methods section.

5. *Line 107-109: what controls were included for metagenomic sequencing? And how did you rule out potential cross-contamination between samples and sequencing libraries?*

Response from authors: Negative controls were not included during sequencing due to cost limitations. However, we ruled out cross-contamination of the samples as Jingmen tick virus was only detected in 3 tick pools from the total 13 tick pools analyzed. Jingmen tick virus was detected only in tick pools of the species *Amblyomma mixtum* collected from hosts (horses) in a single horse paddock in Siberia, Limón. The complete description of viral communities of ticks from these 13 tick pools will be included in a

separate manuscript, but we detected differences in virome composition according to sampling site, sampling method, and tick species, confirming that cross-contamination between samples is unlikely.

6. *Line 108: what criteria were used to filter reads?*

Response from authors: Raw reads obtained from the NextSeq 2000 Illumina platform were processed with the pipeline Microseek (as mentioned in the Materials and Methods Section in the section *Virus taxonomic assignment*). We trimmed raw reads with AlienTrimmer with default options: -k 10 -m 5 -l 50 -p 80 -c 012 -q 20 (5 mismatches allowed, minimum length of 50, quality cutoff of 20, 80% of the read above this cutoff). Read coverage was normalized with BBnorm with options target = 100 min = 1 (target average depth of 100×, minimum depth of 1). Remaining reads were assembled with Megahit with options —min-contig-len 100 (keeping only contigs with a length of 100 nucleotides) to produce contigs. Then reads were mapped back to contigs with Bowtie, while unassociated reads were kept as singletons. Contigs and singletons were translated into proteins with an ad-hoc program that looks for ORFs keeping sequences with a length greater than 15 amino acids. Taxonomic assignment was performed through three steps of blast homology searches. First, with Diamond blastp against the latest RVDB-prot database with a maximum e-value requirement of 5, and annotation is finalized by selecting the common ancestor of all the best matches (LCA annotation). Afterward, sequences were reannotated against the NCBI NR protein database, with a maximum e-value of 100, further annotated with LCA, and sequences that did not belong to viruses were discarded. Finally, previously obtained nucleic sequences are reannotated using blastn against the NCBI NT database. All three steps ensured the elimination of false positive annotations as viral sequences. Additionally, ORFs of 666 amino acids or more that previously lacked matches were tested against the RVDB-prot-hmm database using HMMER with the —domtblout parameter to improve annotation.

For each pool, viral contigs and singletons were mapped onto a JMTV reference genome to produce a draft consensus. Upon alignment of the draft consensus and assessment of high identity (see above), we mapped all contigs and singletons generated from the Microseek pipeline onto a reference genome to produce a draft genome. Finally, trimmed sequence reads from all 3 pools were mapped back onto the draft genome to produce the final genome (MAG).

7. *Line 145-147: Did you perform some kind of recombination test on your sequences, preferably multiple different methods?*

Response from authors: We did not perform intra-segment recombination tests as our purpose was only to identify putative reassortment events by phylogenetic analyses and nucleotide pairwise comparison matrices. As noted by reviewer 1, our results can only hypothesize a reassortment event, not confirm it. Thus, our claims about reassortment were rephrased throughout the manuscript, and when mentioned, it was stated as a hypothesis.

8. *Line 166: "infecting" is doing some heavy lifting here. To my knowledge, evidence of "infection" with Jingmenviruses is scant however, there have been frequent "detections" of Jingmenviruses in multiple different species, mostly using molecular methods.*

Response from authors: The phrase "...infecting various hosts..." was changed to "...detected in various hosts..." in line 176.

Re: Spectrum04078-25R1 (Detection and phylogenetic characterization of Jingmen tick virus in Amblyomma mixtum ticks from Costa Rica)

Dear Ms. Tatiana Murillo:

Your manuscript has been accepted, and I am forwarding it to the ASM production staff for publication. Your paper will first be checked to make sure all elements meet the technical requirements. ASM staff will contact you if anything needs to be revised before copyediting and production can begin. Otherwise, you will be notified when your proofs are ready to be viewed.

Sincerely,
Day-Yu Chao
Editor
Microbiology Spectrum

Reviewer #1 (Comments for the Author):

the authors have addressed the points I raised

Reviewer #2 (Comments for the Author):

The authors have addressed my concerns.